# The Effect of Educational Intervention on Human Papillomavirus Knowledge among Male and Female College Students in Riyadh

**DOI:** 10.3390/medicina60081276

**Published:** 2024-08-07

**Authors:** Esraa Aldawood, Lama Alzamil, Deemah Dabbagh, Taghreed A. Hafiz, Sarah Alharbi, Mohammad A. Alfhili

**Affiliations:** Department of Clinical Laboratories Sciences, The College of Applied Medical Sciences, King Saud University, Riyadh 12372, Saudi Arabia; lalzamil@ksu.edu.sa (L.A.); daldabbagh@ksu.edu.sa (D.D.); thafiz@ksu.edu.sa (T.A.H.); sarralharbi@ksu.edu.sa (S.A.); malfeehily@ksu.edu.sa (M.A.A.)

**Keywords:** HPV, educational, intervention, knowledge

## Abstract

*Background and Objectives*: Persistent high-risk Human Papillomavirus (HPV) can cause cancers in the cervix, vulva, vagina, anus, penis, and oropharynx. A lack of knowledge about HPV can lead to vaccine hesitancy, which is detrimental to combating HPV-related diseases. This study aimed to assess the effectiveness of an HPV educational intervention to enhance university students’ awareness of HPV. *Materials and Methods*: We conducted a quasi-experimental one-group pre-test and post-test study on male and female college students from the College of Applied Medical Science and the College of Nursing in Riyadh, Saudi Arabia, at King Saud University. Data were collected from May 2023 to March 2024. The first section of the survey assessed sociodemographic factors, and the second section measured knowledge regarding HPV. *Results*: A total of 271 students completed the surveys, with 71 males (26.2%) and 200 females (73.8%) participating. Students aged 22 years or older had better HPV awareness. Gender significantly predicts HPV awareness, with female students being four times more likely to be aware of HPV compared to male students. After the educational intervention, significant improvements in HPV knowledge were observed in all items (*p*-values < 0.0001) and across all demographic groups. Misconceptions about HPV were corrected, and the overall knowledge score increased from 29.3% to 82.0%. *Conclusions*: Our results suggest that similar interventions could benefit other populations in the kingdom, potentially increasing vaccination rates.

## 1. Introduction

Cervical cancer is a significant health concern globally, including in Saudi Arabia, where it ranks among the top ten most commonly reported malignancies [1,2,3]. Persistent high-risk Human Papillomavirus (HPV) infections, particularly type 16, are known to cause cancers in various parts of the body, including the cervix, vulva, vagina, anus, penis, and oropharynx [4]. While cervical cancer can be largely prevented through HPV-based screening methods (e.g., Pap smears) and vaccination [5], awareness and knowledge about HPV remain critical in combating HPV-related diseases.

In Saudi Arabia, several studies have highlighted the low levels of HPV awareness among different demographic groups. For example, a study of university students revealed limited knowledge about HPV, particularly among males, who showed a 40% gap in awareness. Additionally, it was found that students in health disciplines exhibited lower awareness than medical students [6]. Similarly, research involving dental practitioners highlighted a gap in knowledge regarding the correlation between HPV and carcinomas [7]. A study conducted among female university students in the United Arab Emirates revealed similar patterns of limited detailed knowledge despite a general awareness of HPV and its link to cervical cancer [8].

Additionally, a study by Aldawood et al. (2023) revealed low awareness and acceptance of the HPV vaccine, with hesitancy rates of 43.9% among females and 75.9% among males. The most common reason for vaccine hesitancy in this study was a lack of knowledge about it [9]. Furthermore, a cross-sectional study on Saudi women of childbearing age revealed that while there is recognition of cervical cancer risk factors, the uptake of the HPV vaccine remains low due to concerns about potential side effects and a lack of comprehensive information [10].

These findings underscore the urgent need for targeted educational interventions to enhance HPV knowledge and reduce vaccine hesitancy. Quasi-experimental studies conducted in Saudi Arabia have shown significant improvements in knowledge and awareness following an educational intervention on other health topics [11,12]. However, there is a noticeable gap in interventions specifically addressing HPV awareness, particularly ones including both male and female university students.

Therefore, this study aims to assess the effectiveness of an HPV educational intervention designed to enhance the awareness and knowledge of male and female university students in Saudi Arabia. This approach is novel in its inclusion of both genders and its focus on a comprehensive educational strategy to address misconceptions, which could potentially improve vaccine uptake.

## 2. Materials and Methods

### 2.1. Study Design and Participants

We performed a quasi-experimental one-group pre-test and post-test study to evaluate the effectiveness of an HPV educational program on male and female college students in Riyadh, Saudi Arabia. We chose to educate college students because this group had never been targeted by the HPV vaccine national immunization schedule, which was only implemented in 2017 at no cost for girls aged 11 or 12. This study was conducted at King Saud University and included students from two colleges: The College of Applied Medical Sciences and the College of Nursing. Participants were recruited using a non-probability convenience sampling technique. Invitations to participate were made during regular class sessions and through departmental communication channels. We emphasized this study’s voluntary nature and ensured that students who agreed to participate could complete both the pre-test and post-test as part of their involvement. A total of 271 college students who completed the pre- and post-tests and consented to participate in this study were included. The inclusion criteria were full-time students registered in an undergraduate program in either the College of Applied Medical Sciences or the College of Nursing and willingness to participate in this study. Students enrolled in the preparatory year or in internships were excluded from this study.

### 2.2. Intervention

Before developing the content for the HPV educational package, an extensive review of the literature on HPV was conducted. Additionally, the official Saudi Ministry of Health website was utilized to structure the content (https://www.moh.gov.sa/en/HealthAwareness/EducationalContent/Diseases/Infectious/Pages/014.aspx, accessed on 1 January 2020). The slides included the following sections: an overview of HPV, risk factors, HPV types, transmission, symptoms, cancers associated with HPV, diagnosis, treatment, and prevention. The teaching method involved a lecture and discussion using PowerPoint slides. The total duration of the HPV educational package was 1 h, primarily consisting of lecture discussions.

### 2.3. Data Collection

We targeted students from the College of Nursing and the College of Applied Medical Sciences at King Saud University. The population of students in the College of Nursing is 1326, and the population of students in the College of Applied Medical Sciences is 1929, resulting in a combined total population of 3255 students. Using Raosoft, Inc. (Seattle, WA, USA) (http://www.raosoft.com/samplesize.html, accessed on 1 January 2020), a 5% margin of error, a 90% confidence level, and an estimated 50% response distribution, the minimum sample size required was calculated to be 250.

A total of 358 students were invited to participate, and 271 consented and completed both the pre-test and post-test, resulting in a response rate of approximately 75.7%. Reasons for non-participation included scheduling conflicts, a lack of interest in the study topic, concurrent academic commitments, and personal preferences. Data were collected from May 2023 to March 2024 using a two-part questionnaire that was validated in a previous study [6]. Data collection occurred on the same day as the intervention, with pre-tests administered before and post-tests conducted immediately after the educational session. This allowed for the immediate assessment of the impact of the educational program. The first section assessed sociodemographic factors, and the second section measured knowledge regarding HPV through 17 single-response true-or-false items that were formulated and validated by Waller et al. (2013) [13]. The content areas of the survey included “General knowledge” (five items), “Infection and complications” (four items), “Risk factors, signs, and symptoms” (three items), and “Mode of spread, prevention, and treatment” (five items). A correct response was scored as one, and an incorrect answer was scored as zero.

### 2.4. Ethical Consideration

Participants were informed of this study’s objectives and invited to voluntarily complete the survey after giving informed consent. This study complied with the Declaration of Helsinki guidelines and received approval from the Institutional Review Board of King Saud University in Riyadh, Saudi Arabia (Ref. No. 23/0183/IRB). All responses were maintained as anonymous and confidential.

### 2.5. Statistical Analysis

Both descriptive and categorical statistics were used whenever appropriate. The frequency and percentage of each demographic data parameter were evaluated, including age, college, nationality, marital status, Grade Point Average (GPA), smoking habits, and history of sexually transmitted diseases. To assess the association of awareness with sociodemographic characteristics, we conducted a logistic regression model using binary awareness (yes vs. no) as the dependent variable, while age group, gender, college, university level, GPA, and smoking status were independent variables, for which we report odds ratios (ORs) and 95% confidence intervals (CIs). For the distribution of the respondents regarding HPV knowledge, percentages were calculated per item, and a Chi-square test was used to calculate the *p*-value. Mean and standard deviation (SD) were calculated for the total knowledge scores, and a two-way ANOVA was used to calculate the *p*-value. For the knowledge score based on sociodemographic characteristics before and after the intervention, mean, standard error, and 95% CI were calculated, and a paired t-test was performed. All data and statistical analyses were performed using GraphPad Prism 10 (GraphPad Software, San Diego, CA, USA).

## 3. Results

### 3.1. Descriptive Characteristics of Students Participating

A total of 271 college students, 71 males (26.2%) and 200 females (73.8%), participated in this study (Table 1). Most students were within the age group of 20–21 years (males: 53.52%; females: 55.5%), and a higher percentage (86.72%) were enrolled in the College of Applied Medical Sciences. A small percentage (13.28%) of the students, who were all female, were from the College of Nursing. Almost all students were Saudi (98.89%) and unmarried (99.63%). Around half (52.39%) of the students had a GPA of 4 or higher, with a higher percentage seen among females (64%). A high percentage of students (96%) were non-smokers. Nearly all students (98.89%) reported that they did not have a history of sexually transmitted diseases, with only three female students reporting that they had such a history. More than half (57.56%) of the students had heard of HPV, with the percentage of female students (65.5%) being higher than the percentage of male students (35.2%).

The roles of several sociodemographic characteristics, including age, gender, university level, GPA, and smoking status, in predicting HPV awareness were evaluated (Table 2). As shown in Table 2, age and gender were identified as significant predictors of HPV awareness among the study participants. There was a strong association between age group and the awareness of HPV. Students aged 22 years or older had better HPV awareness than other age groups (OR = 5.738, 95% CI = 1.604 to 23.24, *p* = 0.0096). Also, gender significantly predicted HPV awareness, as female students were four times more likely to be aware of HPV compared to male students (OR = 4.703, 95% CI = 2.275 to 10.13, *p* < 0.0001). Smokers were more likely to be aware of HPV compared to non-smokers (OR = 6.15, 95% CI = 0.9717 to 121.0, *p* = 0.10), although this result is not statistically significant. Other sociodemographic factors, such as the college of study, university level, and GPA, did not show a strong association with HPV awareness.

### 3.2. HPV Knowledge before and after the Educational Intervention

The effectiveness of the educational intervention in improving HPV knowledge among college students was evaluated by measuring the participants’ HPV knowledge before and after the intervention (Table 3 and Figure 1).

Table 3 shows the distribution of correct and incorrect responses about specific HPV knowledge items before and after the educational intervention. Overall, significant improvements in HPV knowledge were observed in all items after the educational intervention, with *p*-values < 0.0001, indicating the intervention’s effectiveness. The results of the post-interventional survey showed that several misconceptions about HPV were effectively corrected. For example, the proportion of participants that possessed the correct knowledge that HPV is not very rare increased from 26.2% pre-intervention to 80.44% post-intervention. Also, there was a significant increase from 31.37% to 95.20% in the awareness that there are many types of HPV. Furthermore, the number of responses correctly indicating that men can get HPV increased from 37.64% to 93.36%. Similarly, the percentage of students who correctly answered that HPV can cause cervical cancer in females increased from 43.54% to 97.42%, and the knowledge that HPV can cause oropharyngeal cancers in males increased from 12.55% to 71.22%. Other misconceptions regarding HPV, such as that HPV can cause HIV/AIDS or that HPV can be cured with antibiotics, were all corrected after the educational intervention.

The impact of the educational intervention on HPV knowledge was evaluated by calculating the participants’ HPV knowledge scores before and after the intervention across several HPV knowledge domains, including general knowledge, infection and complications, risk factors, signs and symptoms, mode of spread, prevention and treatment, and overall knowledge (Figure 1). Significant improvements in HPV knowledge across all measured domains were observed after the educational intervention. The overall knowledge score increased from 29.3% before the intervention to 82.0% after the intervention, indicating a major improvement in the participants’ understanding of HPV. General knowledge among participants improved from 31.4% before the intervention to 88.7% post-intervention. Also, before the intervention, knowledge about the infection and complications was 25.6%, which increased to 77.8%. Furthermore, participants’ knowledge about the risk factors, signs, and symptoms of HPV increased from 34.1% pre-intervention to 90.2% post-intervention. Similarly, knowledge regarding the mode of spread, prevention, and treatment of HPV improved from 26.1% before the intervention to 71.4% after the intervention.

As shown in Table 4, the improvement in HPV knowledge scores after the educational intervention was statistically significant among all participants across different demographics. The greatest difference in knowledge score before and after the intervention was observed among males (pre-test: 3.13; post-test: 14.04).

## 4. Discussion

This study aimed to assess the effectiveness of an HPV educational intervention among male and female university students. We demonstrated that our educational intervention significantly improved HPV knowledge among students at the colleges of Nursing and Applied Medical Sciences. The pre-intervention responses showed that more than half of the participants had heard of HPV. Factors such as being female and over 22 years old were associated with a higher awareness of HPV, aligning with previous research indicating that older students and females are more likely to be informed about HPV and its risks [6,14].

However, numerous misconceptions about HPV were identified among students before the intervention. These included misinformation on HPV’s prevalence, transmission, and associated health risks. For example, only 26.2% of students correctly knew that HPV is not a rare infection, and a mere 37.64% recognized that males can also contract HPV. A significant majority (87.82%) incorrectly believed that HPV leads to AIDS, and only 29.52% were aware that early sexual activity increases the risk of contracting HPV.

The post-intervention levels of knowledge improved across all areas, with the overall correct response rate nearly tripling from 29.3% to 82.0%. Awareness that HPV infection can cause cervical cancer in women and oropharyngeal cancer in men increased approximately two-fold and six-fold, respectively. These results align with other regional and global studies. For example, an intervention at Princess Nourah Bint Abdulrahman University significantly improved cervical cancer knowledge among students [12]. The program comprised lectures, videos, posters, and interactive sessions across various colleges. After the intervention, recognition of cervical cancer as a preventable disease increased from 50.3% to 76.6%, and awareness of HPV as a risk factor raised from 28.4% to 54.8% [12,15]. Similarly, a study in Guam reported significant improvements in student knowledge of cervical cancer after a 30 min intervention. Awareness that HPV increases cervical cancer risk rose from 72.2% to 99.1%, and recognition of the benefits of regular Pap smears increased from 28.7% to 88.9% [15]. An online educational program for medical and dental trainees in the USA also improved their knowledge of HPV-related cancers. Correct responses about HPV-related cancers increased from 28.4% to 51.9%, and awareness of HPV prevalence rose from 36% to 72% [16]. Another intervention among African American female college students resulted in an improvement in HPV knowledge, as the correct response rate increased from 74% to 91% [17].

In Saudi Arabia, recent studies have highlighted the prevalence of HPV-related misconceptions among college students and the general population [6,9,18,19,20,21,22]. Many of these knowledge gaps were observed among future healthcare providers, such as students attending health and medical colleges. Our study has positive implications for public health by highlighting the importance of educational programs in reducing the burden of HPV-related diseases through improved awareness and prevention strategies. Integrating HPV education into healthcare curricula can significantly empower future professionals and promote informed public health decisions. Targeted educational interventions for university students, especially those in health-related fields, are essential to correct widespread misinformation, leading to better health outcomes and increased HPV vaccination rates.

Our intervention likely succeeded due to several factors. Firstly, it was designed based on a thorough review of the literature and the Saudi Ministry of Health guidelines, ensuring accuracy and relevance. Secondly, the delivery method included a lecture and an active learning component in the form of a discussion, which is believed to promote a deeper understanding of medical topics [23,24]. However, despite the positive outcomes, some misconceptions persisted, such as the belief that HPV can cause AIDS, which a significant number of students (50.55%) continued to hold post-intervention. Therefore, continuous HPV educational efforts are necessary to ensure that students are well informed.

This study had several limitations. Firstly, the knowledge improvement was measured shortly after the intervention, which does not account for long-term retention of knowledge. Secondly, focusing on two colleges within a single university may not accurately represent the broader college student population across the kingdom, indicating a potential sampling bias. Additionally, this study might have been influenced by the Hawthorne effect, where participants’ behaviour may have been altered due to their awareness of being observed. To mitigate this, we ensured anonymity to encourage honest responses, conducted the intervention in a neutral setting to minimize the novelty effect, and standardized procedures for administering pre-tests and post-tests, conducting both tests on the same day to limit potential external influences.

For future studies, it is recommended to assess the long-term retention of HPV knowledge post-intervention. This will help determine the effectiveness of educational programs over time and the need for periodic refreshers. Additionally, expanding this study to include a larger and more diverse sample of students from various universities and regions across Saudi Arabia can help generalize these findings and enhance our understanding of regional differences in HPV awareness. Investigating changes in students’ attitudes and behaviours regarding HPV prevention and vaccination after educational interventions is also essential. Understanding behavioural changes can provide insights into the real-world impact of these programs.

Furthermore, exploring the feasibility and outcomes of integrating comprehensive HPV education into the standard curricula of health and medical colleges is crucial. Assessing the impact of this integration on students’ knowledge and their ability to educate others will provide valuable information. Developing and evaluating community outreach programs led by university students to educate the general public about HPV and evaluating their effectiveness in increasing public awareness is an additional essential recommendation. Additionally, cross-cultural studies comparing the effectiveness of HPV educational interventions in different cultural contexts can help tailor programs to be more culturally relevant and effective. Investigating the use of technological innovations, such as mobile apps or virtual reality, to enhance HPV education and engagement among students and the general population is also recommended.

Implementing these recommendations will enable future research to expand on our findings, leading to enhanced HPV education and prevention strategies.

## 5. Conclusions

In summary, our educational intervention significantly enhanced HPV knowledge among university students in the Nursing and Applied Medical Sciences colleges. It addressed misconceptions and substantially improved students’ understanding of HPV infection. Integrating HPV education into healthcare curricula can empower future professionals and promote informed public health decisions. Sustaining these gains will require continued efforts to shape attitudes toward HPV prevention and vaccination.

## Figures and Tables

**Figure 1 medicina-60-01276-f001:**
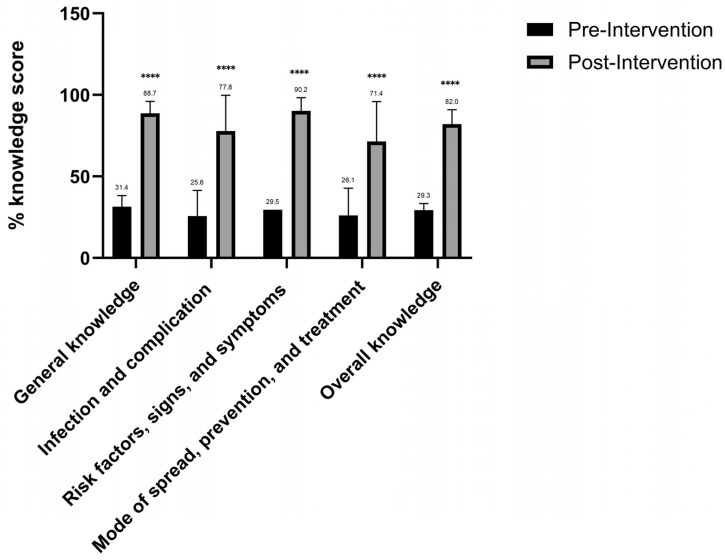
Descriptive analysis of the respondents’ HPV knowledge domains before and after the intervention. Mean and SD shown in bars and error-bars; two-way ANOVA was used as a statistical test. **** represents *p*-value < 0.001.

**Table 1 medicina-60-01276-t001:** Demographic characteristics of study participants.

Item	Male		Female		Total	
	(N = 71)		(N = 200)		(N = 271)	
	N	%	N	%	N	%
Age						
18–19	19	26.76	73	36.5	92	33.95
20–21	38	53.52	111	55.5	149	54.98
22 and older	14	19.72	16	8	30	11.07
In which college are you studying?						
Applied Medical Sciences	71	100	164	82	235	86.72
Nursing	0	0	36	18	36	13.28
Nationality						
Saudi	71	100	197	98.5	268	98.89
Non-Saudi	0	0	3	1.5	3	1.11
Marital Status						
Single	71	100	199	99.5	270	99.63
Married	0	0	1	0.5	1	0.37
GPA						
4 or more	14	19.72	128	64 **	142 *	52.39 **
Less than 4	56	78.87	60	30 **	116 *	42.80 **
Do you smoke?						
Yes	5	7.04	5	2.5	10	3.69
No	66	92.96	195	97.5	261	96.31
Do you have a history of any sexually transmitted disease?						
Yes	0	0	3	1.5	3	1.11
No	71	100	197	98.5	268	98.89
Have you heard of the Human Papillomavirus (HPV)?						
Yes	25	35.21	131	65.5	156	57.56
No	46	64.79	69	34.5	115	42.44

* Total is 258, as 13 participants did not share this information. ** Total does not add up to 100, as 13 participants chose to withhold information.

**Table 2 medicina-60-01276-t002:** Prediction of HPV awareness by sociodemographic characteristics.

Sociodemographic Characteristics	Awareness of HPV		
	OR	95% CI	*p*-Value
Age			
18–19	Ref		
20–21	1.28	0.7045 to 2.343	0.4147
22 or more	5.738	1.604 to 23.24	0.0096
In which college are you studying?			
Applied Medical Sciences	Ref		
Nursing	0.58	0.2568 to 1.305	0.186
What is your university level?			
Year 2	Ref		
Year 3	1.26	0.6633 to 2.413	0.4794
Year 4 or more	1.44	0.6620 to 3.158	0.3633
Gender			
Male	Ref		
Female	4.703	2.275 to 10.13	<0.0001
Cumulative grade point average (GPA)			
4 or more	1.33	0.7225 to 2.449	0.3541
Less than 4	Ref		
Do you smoke?			
Yes	6.15	0.9717 to 121.0	0.10
No	Ref		

OR = odds ratio. CI = confidence interval.

**Table 3 medicina-60-01276-t003:** Distribution of the respondents regarding HPV knowledge.

Items	Pre-Intervention				Post-Intervention				
	Incorrect		Correct		Incorrect		Correct		*p*-Value
	N	%	N	%	N	%	N	%	
Human Papillomavirus (HPV) is very rare	200	73.80	71	26.20	53	19.56	218	80.44	<0.0001
There are many types of HPV	186	68.63	85	31.37	13	4.80	258	95.20	<0.0001
Men cannot get HPV	169	62.36	102	37.64	18	6.64	253	93.36	<0.0001
Most sexually active people will get HPV at some point in their lives	208	76.75	63	23.25	52	19.19	219	80.81	<0.0001
A person could have HPV for many years without knowing it	166	61.25	105	38.75	17	6.27	254	93.73	<0.0001
HPV can cause cervical cancer in females	153	56.46	118	43.54	7	2.58	264	97.42	<0.0001
HPV can cause oropharyngeal cancers in males	237	87.45	34	12.55	78	28.78	193	71.22	<0.0001
HPV can cause HIV/AIDS	238	87.82	33	12.18	137	50.55	134	49.45	<0.0001
HPV can cause genital warts	178	65.68	93	34.32	19	7.01	252	92.99	<0.0001
HPV always has visible signs or symptoms	203	74.91	68	25.09	50	18.45	221	81.55	<0.0001
Having many sexual partners increases the risk of getting HPV	142	52.40	129	47.60	6	2.21	265	97.79	<0.0001
Having sexual intercourse at an early age increases the risk of getting HPV	191	70.48	80	29.52	24	8.86	247	91.14	<0.0001
HPV can be passed on by genital skin-to-skin contact	191	70.48	80	29.52	26	9.59	245	90.41	<0.0001
HPV can be passed on during sexual intercourse	136	50.18	135	49.82	7	2.58	264	97.42	<0.0001
Using condoms reduces the risk of getting HPV	202	74.54	69	25.46	108	39.85	163	60.15	<0.0001
HPV can be cured with antibiotics	210	77.49	61	22.51	73	26.94	198	73.06	<0.0001
HPV usually doesn’t need any treatment	263	97.05	8	2.95	173	63.84	98	36.16	<0.0001

**Table 4 medicina-60-01276-t004:** HPV knowledge scores based on sociodemographic characteristics before and after the intervention.

			Pre-Intervention	Post-Intervention	Post–Pre			
Demographic Characteristic		N	Mean (SE)	Mean (SE)	Mean Difference	95% CI	t-Value (df)	*p*-Value
Gender	Male	71	3.13 (0.56)	14.04 (0.28)	10.92	9.893, 11.94	21 (70)	<0.0001
	Female	200	5.55 (0.35)	13.68 (0.16)	8.13	7.560, 8.700	28 (199)	<0.0001
Age	18–19	92	4.10 (0.52)	13.8 (0.19)	9.674	8.703, 10.65	19.78 (91)	<0.0001
	20–21	149	4.86 (0.41)	13.6 (0.21)	8.758	8.100, 9.417	26.27 (148)	<0.0001
	22 or more	30	7.73 (0.88)	14.6 (0.46)	6.867	5.470, 8.263	10.06 (29)	<0.0001
College	Applied Medical Sciences	235	5.01 (0.33)	14.0 (0.14)	9.03	8.481, 9.579	32 (234)	<0.0001
	Nursing	36	4.33 (0.88)	12.1 (0.42)	7.875	6.153, 9.597	9.325 (31)	<0.0001
University Level	Year 2	102	3.86 (0.48)	13.4 (0.22)	9.51	8.596, 10.42	20.65 (101)	<0.0001
	Year 3	96	4.96 (0.51)	13.9 (0.23)	8.969	8.091, 9.847	20.28 (95)	<0.0001
	Year 4 or more	73	6.34 (0.58)	14.2 (0.31)	7.808	6.962, 8.655	18.39 (72)	<0.0001
GPA	4 or more	142 *	5.94 (0.43)	13.7 (0.21)	7.718	7.041, 8.396	22.52 (141)	<0.0001
	Less than 4	116 *	3.40 (0.42)	13.9 (0.21)	10.47	9.722, 11.23	27.59 (115)	<0.0001
Smoking	Yes	10	8.5 (1.10)	15.1 (0.59)	6.6	4.489, 8.711	7.071 (9)	<0.0001
	No	261	4.78 (0.31)	13.7 (0.15)	8.946	8.417, 9.476	33.26 (260)	<0.0001

* Total is 258, as 13 participants did not share this information, (df) is degree of freedom.

## Data Availability

The data that support the findings of this study are available on request from the corresponding author, E.A.

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
