# Peer review of "The Effect of Educational Intervention on Human Papillomavirus Knowledge among Male and Female College Students in Riyadh"

_medicina, 2024, doi:10.3390/medicina60081276_

Round 1

Reviewer 1 Report

Comments and Suggestions for Authors

Major Comments:

Abstract

1. Authors are suggested to briefly mention the pre-post study design, specific number of colleges and exact p-values in the abstract.

2. Authors should claim their results with significant values.

Introduction

3. Authors are suggested to remove unnecessary information about cervical cancer and HPV.

4. Authors must emphasize the knowledge gap and the study’s specific aim, highlighting its novelty.

Materials and Methods

5. Authors should provide the sample size calculation and justification.

6. Authors are suggested to include the STROBE check list (as an Appendix), and verify if they are complying with all the items on the STROBE check list.

7. What were the inclusion and exclusion criteria? Explain if the questionnaire was pilot tested.

Discussion

8. Authors must include a paragraph describing the public health implications.

9. Authors should describe any limitations of their study including sampling bias and short-term knowledge assessment. Furthermore, these limitations should be discussed.

10. Authors are suggested to discuss implications in detail and outline the specific research recommendations for future.

General comments:

1. Re-check the format. Writing format should be consistent throughout the article.

2. Ensure the addition of updated and latest references.

3. Provide abbreviations used.

Comments on the Quality of English Language

Re-check the format and grammar. Writing format should be consistent throughout the article

Author Response

Major Comments:
Abstract
1. Authors are suggested to briefly mention the pre-post study design, specific number of colleges
and exact p-values in the abstract
2. Authors should claim their results with significant values.
Response to 1 and 2: Thank you for your valuable feedback. We have revised the abstract to
explicitly mention the study design as a quasi-experimental one-group pre-test and post-test study.
We have also specified that the study was conducted with students from the College of Applied
Medical Sciences and the College of Nursing at King Saud University. The exact p-values (<
0.0001) were already included in the original abstract to highlight the significance of the observed
improvements.
Introduction
3. Authors are suggested to remove unnecessary information about cervical cancer and HPV.
Response: Thank you for your suggestion. We have revised the introduction to remove detailed
descriptions of cervical cancer and HPV, focusing instead on the relevance of HPV awareness and
the gaps identified in previous studies.
4. Authors must emphasize the knowledge gap and the study’s specific aim, highlighting its
novelty.
Response: We appreciate your feedback. We have revised the introduction to emphasize the
knowledge gap regarding HPV awareness, particularly among university students in Saudi Arabia.
We have also highlighted the specific aim of our study and its novelty, particularly in including
both male and female students and employing a comprehensive educational intervention.

Materials and Methods
5. Authors should provide the sample size calculation and justification.
Response: We appreciate your valuable suggestion and have included the sample size calculation
and justification in the "Materials and Methods" section. Mentioned in line 229-322
“We targeted students from the College of Nursing and the College of Applied Medical Sciences
at King Saud University. The population of students in the College of Nursing is 1,326 and in the
College of Applied Medical Sciences is 1,929, making a combined total population of 3,255
students. Using Raosoft, Inc. (http://www.raosoft.com/samplesize.html), a 5% margin of error, a
90% confidence level, and an estimated 50% response distribution, the minimum sample size
required was calculated to be 250”
6. Authors are suggested to include the STROBE check list (as an Appendix), and verify if they
are complying with all the items on the STROBE check list.
Response: We appreciate the reviewer’s suggestion regarding the inclusion of the STROBE
checklist. We have verified our manuscript against the STROBE checklist and ensured compliance
with all relevant items. The completed STROBE checklist has been added as an Appendix to the
manuscript.
7. What were the inclusion and exclusion criteria? Explain if the questionnaire was pilot tested.
Response: Thank you for your comments. The inclusion and exclusion criteria for the study have
already been detailed in the manuscript under the section "2.1. Study Design and Participants."
Line 216-219. To restate, the inclusion criteria were full-time students enrolled in an undergraduate
program at either the College of Applied Medical Sciences or the College of Nursing at King Saud
University, who were willing to participate in the study. The exclusion criteria were students
enrolled in the preparatory year or those participating in internships.
The questionnaire used in this study was not pilot tested specifically for this research because it
had already been validated in our previous study:
Aldawood, E.; Alzamil, L.; Faqih, L.; Dabbagh, D.; Alharbi, S.; Hafiz, T.A.; Alshurafa, H.H.;
Altu-khais, W.F.; Dabbagh, R. "Awareness of Human Papillomavirus among Male and Female
University Students in Saudi Arabia," Healthcare (Switzerland) 2023, 11, 1–12,
doi:10.3390/healthcare11050649.
The validation and testing performed in this earlier study ensured the reliability and validity of the
questionnaire, which we have used without modification in the current research. This information
was mentioned in line 327 of the manuscript.

Discussion
8. Authors must include a paragraph describing the public health implications.
Response: Thank you for your valuable feedback. We have revised the manuscript to include a
paragraph in the discussion describing the public health implications of our study in line 492 to
498
9. Authors should describe any limitations of their study including sampling bias and short-term
knowledge assessment. Furthermore, these limitations should be discussed.
Response: Thank you for your insightful comments. We have addressed these limitations in the
discussion in line 508 to 707.
10. Authors are suggested to discuss implications in detail and outline the specific research
recommendations for future.
Response: Thank you for your valuable feedback. We have revised the manuscript to include a
paragraph describing the implications line 492 to 498 and the recommended future studies in line
708 to 727

General comments:
1. Re-check the format. Writing format should be consistent throughout the article.
Response: Thank you for your comment. Corrected to using UK spelling throughout the paper.
2. Ensure the addition of updated and latest references.
Response: Thank you for your comment regarding the addition of updated and latest references.
We have ensured that the references used in our manuscript are the most recent and relevant to our
study
For instance:
Alfhaid, F.; , Mansour Khater Alzahran , Mohammed Zaid Aljulifi , Yousef Alrohaimi, Maram
Nasser Alawlah3 , Fatimah Lailay M. AlMutairi4, Sara Mohammad H. Alkahtani5, Moudi
Abdulrahman Al-mousa6, S.N.A. 1 Prevalence and Perception of HPV Vaccination Among Health Science Students in Saudi Arabia. Journal of Pharmacy and Bioallied Sciences | 2024, 7, 1–5,
doi:10.4103/jpbs.JPBS.
Alshammari, F.; Khan, K.U.; Altamimi, T.; Lingam, A.S.; Koppolu, P.; Alhussein, S.A.;
Abdelrahim, R.K.; Abusalim, G.S.; ElHaddad, S.; Asrar, S.; et al. Knowledge, Attitudes and
Perceptions Regarding Human Papillomavirus among University Students in Hail, Saudi Arabia.
Asian Pac J Cancer Prev 2023, 16, 1–14, doi:10.7717/peerj.13140.
Aldawood, E.; Alzamil, L.; Faqih, L.; Dabbagh, D.; Alharbi, S.; Hafiz, T.A.; Alshurafa, H.H.;
Altukhais, W.F.; Dabbagh, R. Awareness of Human Papillomavirus among Male and Female
University Students in Saudi Arabia. Healthcare (Switzerland) 2023, 11, 1–12,
doi:10.3390/healthcare11050649.
Gari, A.; Ghazzawi, M.A.; Ghazzawi, S.A.; Alharthi, S.M.; Yanksar, E.A.; Almontashri, R.M.;
Batarfi, R.; Kinkar, L.I.; Baradwan, S. Knowledge about Cervical Cancer Risk Factors and Human
Papilloma Virus Vaccine among Saudi Women of Childbearing Age: A Community-Based Cross-
Sectional Study from Saudi Arabia. Vaccine X 2023, 15, 100361,
doi:10.1016/j.jvacx.2023.100361.
Thanasuwat, B.; Leung, S.O.A.; Welch, K.; Duffey-Lind, E.; Pena, N.; Feldman, S.; Villa, A.
Human Papillomavirus (HPV) Education and Knowledge Among Medical and Dental Trainees. J
Cancer Educ 2023, 38, 971–976, doi:10.1007/s13187-022-02215-2.
Almaghlouth, A.K.; Bohamad, A.H.; Alabbad, R.Y.; Alghanim, J.H.; Danah, J. Acceptance,
Awareness, and Knowledge of Human Papillomavirus Vaccine in Eastern Province, Saudi Arabia.
2022, 14, doi:10.7759/cureus.31809.
Lingam, A.S.; Koppolu, P.; Alhussein, S.A.; Abdelrahim, R.K.; Abusalim, G.S.; Elhaddad, S.;
Asrar, S.; Nassani, M.Z.; Gaafar, S.S.; Bukhary, F.M.T.; et al. Dental Students’ Perception,
Awareness and Knowledge About HPV Infection, Vaccine, and Its Association with Oral Cancer:
A Multinational Study. Infect Drug Resist 2022, 15, 3711–3724, doi:10.2147/IDR.S365715.
Somera, L. P., Diaz, T. P., Mummert, A., Choi, J., Ayson, K., & Badowski, G. Cervical Cancer
and HPV Knowledge and Awareness: An Educational Intervention among College Students in
Guam. Cancer Epidemiology Biomarkers & Prevention 2022, 31.
3. Provide abbreviations used.
Response: Thank you for your valuable feedback. We have ensured that all abbreviations and
acronyms are defined at their first mention in both the abstract and the main text.

Reviewer 2 Report

Comments and Suggestions for Authors

Thank you for the opportunity to review manuscript ID: medicina-3110164. This study aimed to assess the effectiveness of an HPV educational intervention to enhance university students' awareness about HPV in Riyadh, Saudi Arabia.

General impression:  

The article is well organized and written.

Comments of the article:   

  • You should add relevant dates to the abstract.
  • It is necessary to provide a detailed description of the study setting.
  • It is necessary to provide a detailed description of the method of recruiting participants for the study.
  • State the reasons for non-participation of a significant percentage of students.
  • It is necessary to provide a detailed description of the relevant dates, including the periods of recruitment, exposure, follow-up and data collection.
  • It is necessary to provide a detailed description of the time that passed from the pre-test to the intervention, as well as from the intervention to the post-test.
  • Please elucidate the methodology employed to address the missing data, given that the questionnaires were distributed to students in a classroom setting. It is highly probable that the missing data is present within this extensive sample.
  • In the Discussion section, there is less discussion about data related to HPV, and a lot of discussion about the HPV vaccine - which was not even included in the questionnaire. Correct this.
  • I suggest that the authors include a thorough analysis of the limitations imposed by social desirability bias, the Hawthorne effect. It is important to acknowledge that the presence of classmates, teachers and the overall classroom environment can have a significant impact on responses, either positively or negatively.
  • Could you please describe any efforts to address potential sources of bias?

Author Response

You should add relevant dates to the abstract.
Response: Thank you for pointing this out. We have added the relevant dates for data collection
to the abstract, specifying that data were collected from May 2023 to March 2024.
• It is necessary to provide a detailed description of the study setting.
Response: Thank you for your valuable feedback. According to STOPE guidelines setting
includes the following and all were detailed in our study:
Study Setting and Locations: The study was conducted at King Saud University, located in
Riyadh, Saudi Arabia. It involved students from two specific colleges within the university: The
College of Applied Medical Sciences and the College of Nursing. Mentioned in line 229.
Relevant Dates:
• Recruitment Period: May 2023, when students were invited to participate in the study
until March 2024. Mentioned in line 326-327
• Exposure/Intervention Period: This intervention involved a 1-hour lecture and
discussion using PowerPoint slides. Mentioned in line 226-227.
• Data Collection: Data collection occurred on the same day as the intervention, with pretests
administered before and post-tests conducted immediately after the educational
session. This allowed for the immediate assessment of the impact of the educational
program. Mentioned in line 327-330.
• It is necessary to provide a detailed description of the method of recruiting participants for the
study.
Response: Thank you for your valuable feedback. Participants were recruited using a nonprobability
convenience sampling technique. Invitations to participate were made during regular
class sessions and through departmental communication channels. We emphasized the study’s
voluntary nature and ensured that students who agreed to participate could complete both the pretest
and post-test as part of their involvement. Mentioned in line 210-214.
• State the reasons for non-participation of a significant percentage of students.
Response: Thank you for highlighting the need to clarify the reasons for non-participation. We
have added a detailed explanation of the non-participation reasons to the manuscript. The revised
section now reads:

A total of 358 students were invited to participate, and 271 consented and completed both the pretest
and post-test, resulting in a response rate of approximately 75.7%. Reasons for nonparticipation
included scheduling conflicts, lack of interest in the study topic, concurrent academic
commitments and personal preferences. Line 323-326.

It is necessary to provide a detailed description of the relevant dates, including the periods of
recruitment, exposure, follow-up and data collection.
Response: Thank you for your valuable comment. As described earlier:
Relevant Dates:
• Recruitment Period: May 2023, when students were invited to participate in the study
until March 2024. Mentioned in line 326-327
• Exposure/Intervention Period: This intervention involved a 1-hour lecture and
discussion using PowerPoint slides. Mentioned in line 226-227
• Data Collection: Data collection occurred on the same day as the intervention, with pretests
administered before and post-tests conducted immediately after the educational
session. This allowed for the immediate assessment of the impact of the educational
program. Mentioned in line 327-330.
• It is necessary to provide a detailed description of the time that passed from the pre-test to the
intervention, as well as from the intervention to the post-test.
Response: Thank you for your valuable feedback. As mentioned earlier
Data collection occurred on the same day as the intervention, with pre-tests administered
before and post-tests conducted immediately after the educational session. This allowed
for the immediate assessment of the impact of the educational program. Mentioned in line
327-330.
• Please elucidate the methodology employed to address the missing data, given that the
questionnaires were distributed to students in a classroom setting. It is highly probable that the
missing data is present within this extensive sample.
Response: Thank you for your valuable feedback. During the HPV awareness intervention study,
some data were missing, primarily because some students chose not to share their GPA, despite
the survey being anonymous. To address this issue, we documented the exact number of missing
responses in the relevant tables.
• In the Discussion section, there is less discussion about data related to HPV, and a lot of discussion
about the HPV vaccine - which was not even included in the questionnaire. Correct this.

Response: Thank you for your appreciated feedback. We have revised the discussion section to
better align with the data related to HPV knowledge.
• I suggest that the authors include a thorough analysis of the limitations imposed by social
desirability bias, the Hawthorne effect. It is important to acknowledge that the presence of
classmates, teachers and the overall classroom environment can have a significant impact on
responses, either positively or negatively. Could you please describe any efforts to address
potential sources of bias?
Response: Thank you for your insightful feedback on our manuscript. We have included a
discussion of how these factors might have affected the responses and the measures taken to
mitigate their impact. Line 702-707
In the revised manuscript, we have described our efforts to address potential sources of bias. This
includes:
• Ensuring anonymity and confidentiality.
• Conducting the intervention in a controlled setting to reduce the Hawthorne effect.
• Implementing a standardized procedure for administering the pre-tests and post-tests to
maintain consistency.
Comments on the Quality of English Language
Re-check the format and grammar. Writing format should be consistent throughout the article
Response: Thank you for pointing out this. We have thoroughly re-checked the manuscript to
ensure that the writing format is uniform throughout the article. We corrected any grammatical
errors and made sure that UK spelling is used consistently.

Reviewer 3 Report

Comments and Suggestions for Authors

The manuscript is a very interesting and useful topic, and the authors present it clearly and thoroughly. My only comment is that as the male participants are not as many as the female ones, on page 4, last sentence of the first paragraphs, I recommend that the authors instead of writing numbers should be referring to the percentage of males and females accordingly. 

Author Response

The manuscript is a very interesting and useful topic, and the authors present it clearly and
thoroughly.
My only comment is that as the male participants are not as many as the female ones, on page 4,
last sentence of the first paragraphs,
I recommend that the authors instead of writing numbers should be referring to the percentage of
males and females accordingly.

Response: Thank you for your appreciated comment. We have revised the sentence to refer to the
percentage of male and female participants rather than absolute numbers. Line 373-374 “More
than half (57.56%) of students had heard of HPV, with the percentage of female students (65.5%)
higher than the percentage of male students (35.2%)”.

Round 2

Reviewer 1 Report

Comments and Suggestions for Authors

Authors addressed almost all the comments.

Reviewer 2 Report

Comments and Suggestions for Authors

Thank you for the opportunity to re-review manuscript ID: medicina-3110164. The authors have addressed all my comments and made appropriate changes in the revised version of this paper. I thank the authors for their efforts to improve this manuscript.